# Senkyunolide I: A Review of Its Phytochemistry, Pharmacology, Pharmacokinetics, and Drug-Likeness

**DOI:** 10.3390/molecules28083636

**Published:** 2023-04-21

**Authors:** Yan Huang, Yan Wu, Hongxiang Yin, Leilei Du, Chu Chen

**Affiliations:** 1School of Pharmacy, Chengdu University of Traditional Chinese Medicine, Chengdu 611137, China; 2Sichuan Provincial Key Laboratory of Quality and Innovation Research of Chinese Materia Medica, Sichuan Academy of Chinese Medicine Sciences, Chengdu 610041, China; 3School of Ethnic Medicine, Chengdu University of Traditional Chinese Medicine, Chengdu 611137, China

**Keywords:** Senkyunolide I, phthalide, phytochemistry, pharmacology, pharmacokinetics, drug-likeness

## Abstract

Senkyunolide I (SI) is a natural phthalide that has drawn increasing interest for its potential as a cardio-cerebral vascular drug candidate. In this paper, the botanical sources, phytochemical characteristics, chemical and biological transformations, pharmacological and pharmacokinetic properties, and drug-likeness of SI are reviewed through a comprehensive literature survey, in order to provide support for its further research and applications. In general, SI is mainly distributed in Umbelliferae plants, and it is relatively stable to heat, acid, and oxygen, with good blood–brain barrier (BBB) permeability. Substantial studies have established reliable methods for the isolation, purification, and content determination of SI. Its pharmacological effects include analgesic, anti-inflammatory, antioxidant, anti-thrombotic, anti-tumor effects, alleviating ischemia–reperfusion injury, etc. Pharmacokinetic parameters indicate that its metabolic pathway is mainly phase Ⅱ metabolism, and it is rapidly absorbed in vivo and widely distributed in the kidneys, liver, and lungs.

## 1. Introduction

Phthalides are a group of structural specific constituents naturally distributed in several important medicinal herbs in Asia, Europe, and North Africa [1]. Accumulating evidence demonstrated that natural phthalides have various pharmacological activities, including analgesic [2], anti-inflammatory [3], antithrombotic [4], and antiplatelet [5] activities, mostly consistent with the traditional medicinal uses of their natural plant sources. For example, *Ligusticum chuanxiong* Hort. (*L. chuanxiong*) and *Angelica sinensis* (Oliv.) Diels (*A. sinensis*), frequently used in traditional Chinese medicine (TCM) to invigorate the circulation of qi and the blood, both contain a high level of phthalide components, typically exceeding 1% in their rhizome or root [6,7]. One of these phthalides that has been broadly studied is ligustilide (LIG) (Figure 1a), which displays analgesic, anti-inflammatory, antihypertensive, and neuroprotective activities on brain injury [8]. However, LIG is not a promising drug candidate due to its instability, potent lipophilicity, poor water solubility, and low bioavailability. Druggability improvement was achieved, to a certain degree, by preparing LIG into a nano-emulsion or a hydroxypropyl-β-cyclodextrin complex [9,10], but a specific technique is required and the manufacture cost is high. N-butylphthalide (NBP), first isolated from celery seed, has been licensed in China for the indication of mild and moderate acute ischemic stroke [11], and clinical trials of its effects on vascular cognitive disorder as well as amyotrophic lateral sclerosis are ongoing [12]. Still, extensive application of NBP is limited, owing to its hepatotoxicity, poor solubility, and unsatisfactory bioavailability [13]. Therefore, discovering natural phthalides with improved druggability from traditional medicinal herbs is both intriguing and meaningful.

SI (Figure 1b) is also a natural phthalide existing in *L. chuanxiong* and *A. sinensis* in relatively low contents and is generally considered as an oxidation product of LIG. It has similar pharmacological activities but significantly superior stability, solubility, safety, bioavailability, and brain accessibility compared with LIG, thus meriting further druggability research and evaluation. In this paper, the physicochemical characteristics, isolation and purification methods, as well as pharmacological and pharmacokinetic properties of SI are overviewed. An illustrated summary is described in Figure 2.

## 2. Distribution and Production

### 2.1. Distribution in Nature

SI was firstly discovered as a natural phthalide from *Ligusticum wallichii* Franch in 1983, under the name (Z)-ligustidiol [14]. Subsequently, SI was found in the rhizome of *Cnidium officinale* Makino in 1984 [15]. According to the published literature to date, SI was found mainly in Umbelliferae plants, including *Angelica sinensis* (Oliv.) Diels [16], *Ligusticum chuanxiong* Hort [17], *Lomatium californicum* (Nutt) [18], *Cryptotaenia japonica* Hassk [19], and so on.

In general, natural phthalides are distributed mainly in plants belonging to the Umbelliferae family, and are also occasionally found in Cactaceae, Compositae, Lamiaceae, Gentianaceae, and Loganiaceae families. In addition, natural phthalides obtained as fungal and lichen metabolites have been reported [3].

### 2.2. Production

#### 2.2.1. Chemical Transformation from LIG

Only a trace amount of SI can be found in fresh rhizomes of *L. chuanxiong*, while more SI is produced by the degradation of LIG during processing and storage. Li and colleagues [20] investigated the chemical changes induced by different processing methods, and the results indicated that the main phthalides in rhizomes of *L. chuanxiong*, such as LIG and senkyunolide A (SA), decreased significantly. Meanwhile, levistolide A, SI, and its isomer senkyunolide H (SH) increased correspondingly. According to the report, the highest level of SI (0.32 mg/g) was found when fresh rhizomes of *L. chuanxiong* were dried at 60 °C for 24 h. In addition, the chemical changes of rhizomes of *L. chuanxiong* during storage were assayed. The results showed that the contents of LIG, coniferyl ferulate, and SA decreased significantly after 2 years of storage at room temperature, resulting in increases in the quantities of SI, SH, ferulic acid, levistilide A, and vanillin. SI increased by 37.6% during the period of storage and was presumed as the dominating oxidative product of LIG [21].

Duric and co-workers [22] found that LIG is relatively stable in plant oil cells. However, the purified LIG became very unstable and inclined to form dimers or trimers under light, whereas when heated in the dark, it mainly transformed into SI and its isomer SH [23]. The results above are consistent with those reported by Lin et al. [24].

Duan and colleagues [25] studied the reaction products of LIG in an electrochemical reactor. Five products were separated and identified, including two dihydroxyl products SI and SH, as well as an epoxide 6,7-epoxyligustilide. The latter is a key intermediate in the transformation of LIG into SI and SH.

Processing conditions influence SI production in the rhizome of *L. chuanxiong*. A steaming process with or without rice wine resulted in higher SI levels compared to a stir-frying process. [7]. A simple mechanism for the transformation of LIG to SI is illustrated in Figure 3.

#### 2.2.2. Metabolic Transformation of LIG

SI is the major metabolite of LIG in vivo and in vitro. Yan et al. [26] found that SI was one of the main metabolites when LIG was injected intravenously in SD rats. Similarly, LIG can be transformed into SI when incubated with small intestinal homogenates or liver microsomes of rats [25]. When incubating LIG with human or rat hepatocytes at 37 °C, SI was found to be the main metabolite, with proportions of 42% and 70%, respectively [27]. Furthermore, research on the enzyme kinetics of LIG incubated with rat liver microsomes demonstrated that CYP3A4, CYP2C9, and CYP1A2 are the main metabolic enzymes involved in the LIG metabolism [28]. However, the key enzyme catalyzing LIG into SI in vivo has not been identified.

## 3. Chemical and Biological Properties

Pure SI is yellowish amorphous powder or sticky oil with a celery-like smell. Unlike most natural phthalides, SI is soluble in water and some organic solvents, such as ethanol, ethyl acetate, and chloroform. Several studies suggested that SI has better drug-like properties compared to LIG.

### 3.1. Stability

The degradation of SI in aqueous solution conforms to first-order degradation kinetics, and the energy of activation (Ea) was 194.86 k J/mol. SI in weakly acidic solution showed a better stability, while its degradation ratio accelerated significantly under alkalescent conditions [29].

It was reported that oxygen is the dominating factor that accelerates the degradation rates of SI and SA induced by light and temperature. At room temperature with daylight, SA was completely converted into butylphthalide within 2 months, while only about 20% of SI was converted into its cis-trans isomer after 5 months of storage, indicating that SI is more stable than SA [30].

Peihua Zhang et al. [31] introduced a methanol extract of *L. chuanxiong* in boiling water and evaluated the content changes during decoction. As a result, the content of LIG decreased from 14 mg/g to 0.4 mg/g after 20 min, while the SI content increased from 1.4 mg/g to 1.7 mg/g during 60 min of heating.

Formula granules are a type of dried decoction of a prepared herbal medicine. In the characteristic chromatograms of both *A. sinensis* and *L. chuanxiong* formula granules issued by the National Pharmacopoeia Commission of China, SI is marked as a dominant and characteristic peak, suggesting that SI is stable during decoction, concentration, and drying processes. On the contrary, as the most abundant phthalide both in *L. chuanxiong* and *A. sinensis* slices, LIG is almost undetectable in these formula granules [32].

### 3.2. Permeability

SI has satisfactory permeability and solubility. Yuan and co-workers [33] screened the potential transitional components in *L. chuanxiong* extract by a serum pharmacochemical method and high-performance liquid chromatography-diode array detection tandem mass spectrometry/mass spectrometry (HPLC-DAD-MS/MS) analysis. SI was identified as a transitional component both in the plasma and cerebrospinal fluid, while ferulic acid was detected only in plasma. SI can pass through the BBB easily, and the AUC of SI in the brain accounted for 77.9% of that in plasma [34,35].

The water solubility of SI was measured to be 34.3 mg/mL, and the lipid–water partition coefficient was 13.43 [36]. Previous studies revealed that SI exhibits good absorption in the rat gastrointestinal tract, including the jejunum, colon, ileum, and duodenum, and no significant differences in the absorption rate constant and apparent absorption coefficient were observed [37,38].

## 4. Quantitative Analysis

### 4.1. Analytical Methods

The reported analytical methods of SI in herbs and prescriptions, as well as corresponding parameters, are shown in Table 1. The SI analyses were generally performed by high-performance liquid chromatography (HPLC) combined with an ultraviolet (UV) or diode array detection (DAD) detector. Most of the separations were carried out on a C18 column using a mixture of acetonitrile and acidic aqueous solution as the mobile phase. In addition, other detective devices, such as electrospray ionization tandem mass spectrometry (ESI-MS) and time-of-flight mass spectrometry (TOF-MS), were used for the structure elucidation and metabolite analysis of SI.

### 4.2. Content in Medicinal Material and Preparation

The contents of SI medicinal materials and preparations are shown in Table 2 and Table 3, respectively.

Among the commonly used TCM, SI exists limitedly in *A. sinensis* and *L. chuanxiong.* Table 2 indicates that the maximum content of SI in *A. sinensis* is 1 mg/g, while it reaches more than 10 mg/g in *L. chuanxiong*. The reason is presumably that LIG in *L. chuanxiong* is in a higher level and may produce more SI compared with that in *A. sinensis*. In addition, SI concentrations in Chuanxiong dispensing granules range from 2.08 to 6.07 mg/g. The relatively high content might be attributed to its good water solubility or accelerated transformation from LIG during decocting, concentrating, or drying processes. Table 3 summarizes the quantitative analysis of SI in multiple compound preparations containing *L*. *chuanxiong* rhizome and *A. sinensis* root. The results show a large fluctuation from 0.02 to 2.206 mg/g, suggesting that SI content is most likely influenced by material quality, formulation, and preparation technology.

## 5. Extraction and Isolation

The rhizomes of *L. chuanxiong* and roots of *A. sinensis* are commonly used materials for SI extraction, isolation, and purification. Ethanol of high concentration was the most used extraction solvent, followed by methanol and water. Besides the conventional extraction methods, such as reflux, immersion, and ultrasonication, supercritical fluid extraction or ultra-high pressure ultrasonic-assisted extraction was carried out to improve the effect and efficiency.

SI separation and purification were mainly performed by different column chromatographic methods, including flash column chromatography, counter-current chromatography, borate gel affinity column chromatography, and preparative HPLC. The packing materials used were silica gel, RP-C18, and macroporous resin. The details of SI extraction and isolation are shown in Table 4.

**Table 4 molecules-28-03636-t004:** Extraction and isolation of SI.

Raw Material	Extraction Solvent	Extraction Method	Separation Method	Raw Material Consumption (kg)	SI Obtained (g)	Ref.
*Cnidium officinale* rhizome	Hexane, diethyl ether, methanol	\	Repeated elution and purification by silica gel column chromatography	2	0.239	[63]
*Cryptotaenia japonica* herba	75% Ethanol	Cold immersion, ultrasound	Repeated elution of crude fractions by silica gel column chromatography and purification by gel column chromatography	2	0.020	[19]
*Angelica sinensis* root	Methanol	Cold immersion	Rapid silica column chromatography for crude fractionation and preparative RP-18MPC fractionation and preparative HPLC purification	8	0.0038	[16]
*Ligusticum sinense* aerial part	95% Ethanol	\	Repeated elution purification by flash column chromatography and silica gel column chromatography	70	0.010	[64]
*Ligusticum chuanxiong* rhizome	90% Ethanol	Cold immersion, ultrasound	Counter current chromatography	0.03	0.0064	[39]
*Ligusticum chuanxiong* rhizome	Water	Soaking in 80 °C hot water for 1 h	Silica gel column chromatography for crude fractionation and borate affinity gel column chromatography for purification	1	0.791	[65]
*Ligusticum chuanxiong* rhizome	80% Ethanol	Reflux	D-101 macroporous adsorption resin crude fraction and purification by silica gel column chromatography and reverse high-performance preparative liquid chromatography	3	0.962	[66]
*Ligusticum chuanxiong* rhizome	70% Ethanol	Reflux	Crude fractionation of HPD-100 macroporous resin and purification by reversed-phase high-performance liquid chromatography	0.2	0.217	[67]

## 6. Pharmacology

The reported pharmacological activities of SI were summarized in Figure 4 and Table 5.

### 6.1. Protection of the Brain

#### 6.1.1. Neuroprotection of Cerebral Ischemia/Hemorrhage

Due to the high risks of disability and mortality, cerebral hemorrhage and ischemia remain intractable diseases, resulting in neurologic impairment, tissue necrosis, cell apoptosis, and subsequent complications [68]. Previous studies demonstrated that SI performs significant neuroprotection mainly through antioxidant and anti-apoptotic pathways. Hu et al. [69] investigated the protective effect and possible mechanism of SI (36 and 72 mg/kg, i.v.) on cerebral ischemia–reperfusion (I/R) impairment using the rat transient middle cerebral artery occlusion (tMCAO) model. The results indicated that SI could ameliorate neurological injury, reduce cerebral infarct volume, decrease the malonaldehyde (MDA) content, and increase the superoxide dismutase (SOD) activity of brain tissue. The mechanism involves promoting the expression of p-Erk1/2/t-Erk1/2, c-Nrf2, n-Nrf2, HO-1, and NQO1, and deregulating the expression of Bcl-2, Bax, caspase 3, and caspase 9.

The protective effects of compounds (SI, SH, SA, LIG, and ferulic acid) isolated from *L. chuanxiong* were evaluated on an oxygen–glucose deprivation–reoxygenation (OGD/R) model using cultured SH-SY5Y cells. The results demonstrated that both SI and LIG could improve cell viability, reduce reactive oxygen species (ROS), and lactate dehydrogenase (LDH) levels. SI showed a more potent inhibiting activity on LDH compared to LIG [70].

LIG and its metabolites SI and SH have protective effects on the intracerebral hemorrhage (ICH) model caused by autologous blood injection into CD-1 mice. SI could ameliorate neurological deficit, brain edema, and neuronal injury; alleviate microglia cell and astrocyte activations; and reduce peripheral immune cell infiltration caused by ICH. However, SI is less effective than SH. Inhibition of the Prx1/TLR4/NF-κB signal pathway and anti-neuroinflammatory injury are involved in the potential mechanism of LIG and SH [71].

#### 6.1.2. Protection against Septic Encephalopathy

Sepsis is a systemic inflammatory response syndrome caused by microbial infection. Septic encephalopathy (SE) with cerebrovascular dysfunction and neuron growth inhibition is a common complication. SI (36 and 144 mg/kg, i.p.) ameliorates injury on SE rats by increasing Ngb expression, upregulating the p38 MAPK signal pathway, and the consequent promotion of neuron growth [72].

Sleep quality impairment of sepsis rats would accelerate inflammatory factor release, and the prognosis of sepsis may benefit from sleep improvement [71]. SI demonstrated sleep-improving sedative effects, but its role in sepsis is unclear. Thus, a cecal ligation and puncture (CLP)-induced sepsis model using C57BL/6J mice was established. The results showed that SI (36 mg/kg, i.p.) improved the survival rate and cognitive dysfunction of sepsis mice, ameliorated systemic inflammatory response, reduced apoptotic cells in the hippocampus, and inhibited the inflammatory signaling pathway. Surprisingly, the hypothesis that alleviating sleep deprivation could ameliorate SE injury was further confirmed by reversing the expression of sleep deprivation-related markers BNDF and c-FOS after SI administration [73].

### 6.2. Protection of the Liver, Kidneys, and Lungs

Blood supply is critical for ameliorating tissue and organ damage caused by persistent ischemia. SI can attenuate hepatic and renal I/R injury through antioxidant, anti-inflammatory, and anti-apoptotic effects. SI (50, 100, and 200 mg/kg) was injected intraperitoneally to the modified liver I/R murine model. As a result, SI (200 mg/kg) decreased TNF-α, IL-1β, and IL-6 in serum; inhibited the phosphorylation of p65 NF-κB and MAPK kinases; and reduced the expression of Bax and Bcl-2. Furthermore, SI can alleviate H_2_O_2_-induced oxidative damage in HuCCT1 cells, promote the nuclear translocation of Nrf-2, and reduce the levels of ROS and MDA [74]. Administration on renal I/R injury mice confirmed that SI can protect renal function and structural integrity, reverse increases in ischemia–induced blood urea nitrogen (BUN) and serum creatinine (SCr), ameliorate pathological renal damage, and inhibit TNF-α and IL-6 secretions. Furthermore, reductions in ROS production as well as endoplasmic reticulum stress-related protein expressions are involved in the potential protection mechanism [75].

It was reported that SI (36 mg/kg, i.p.) could ameliorate sepsis-related lung injury on cecal ligation and puncture-induced sepsis C57BL/6 mice. SI performed its effects by decreasing protein levels and neutrophil infiltration, inhibiting the phosphorylation of JNK, ERK, P38, and p65, and downregulating TNF-α, IL-1β, and IL-6 in plasma and lung tissue. CD42d/GP5 staining results indicated that platelet activation was decreased after SI administration. Moreover, SI could significantly reduce MPO-DNA levels stimulated by phorbol 12-myristate 13-acetate (PMA) [76].

### 6.3. Protection of Blood and Vascular Systems

#### 6.3.1. Effects on the Blood System

The rhizome of *L. chuanxiong*, a herb commonly used to promote blood circulation and remove blood clots, has drawn interest due to its anticoagulant and antiplatelet activities. Anticoagulant activity was screened by measuring the binding rates between components from herbal extracts and thrombin (THR) in vitro. Preliminary results showed that SI and isochlorogenic acid C could inhibit the activity of THR. The results of molecular docking revealed that SI and isochlorogenic acid C could bind to the catalytic active site of THR [77]. Similarly, *L. chuanxiong* extracts were screened for their possible inhibitory effects on THR and Factor Xa (FXa) using an on-line dual-enzyme immobilization microreactor based on capillary electrophoresis. SI, SA, LIG, and ferulic acid exhibited vigorous THR inhibitory activities, while isochlorogenic acid A could effectively inhibit FXa activity [46].

A study eliminated SI from Siwu decoction (SWD) to explore its contribution to the antiplatelet and anticoagulant activities of the formula. The absence of SI resulted in a significantly shortened activated partial thromboplastin time of SWD, while the active sequence of prothrombin time (PT) was inhibited, indicating that SI plays an important role in the activities of SWD [78].

#### 6.3.2. Effects on the Vascular System

SI can promote angiogenesis and it represents vasodilating and antithrombotic effects, thereby providing protection to the vascular system. SI in Guanxinning tablets could ameliorate endogenous thrombus injury in zebrafish through various pathways, including oxidative stress, platelet activation, and coagulation cascade [79]. In addition, it was reported that SI prevents microthrombus formation by attenuating Con A-induced erythrocyte metamorphic damage and reducing erythrocyte aggregation [80].

Suxiao Jiuxin Pill (SX) is a Chinese patent medicine containing extracts of *L. chuanxiong* and is usually used for coronary heart disease treatment. The potential active components of SX were screened for cell Ca^2+^ regulation activity, which is critical for vascular resistance and pressure handing. SI isolated from SX can amplify cardiovascular diastolic activity through calcium antagonistic activity [81].

Additionally, a study on the effect on the endothelial vascular cell model confirmed that SI might promote the formation of the luminal structure of human microvascular endothelial cells and induce endothelial angiogenesis by upregulating placental growth factor [82].

### 6.4. Other Pharmacological Effects

The analgesic effect of SI was evaluated by an acetic acid-induced writhing test on Kunming mice (8, 16, and 32 mg/kg, i.g.), and the anti-migraine activity was tested by nitroglycerin-induced headaches in SD rats (18, 36, and 72 mg/kg, i.g.). SI (32 mg/kg) significantly elevated the pain thresholds and the number of acetic acid-induced writhing reactions in mice. SI (72 mg/kg) in rats remarkably reduced the NO levels in plasma and brain tissue and increased 5-HT levels in plasma [83]. In another study where rats were dosed with SI (144, 72, and 36 mg/kg, i.p.) to cure the cortical spread of migraine, plasma NO and calcitonin gene-related peptide (CGRP) significantly decreased after SI (144 mg/kg) treatment [84].

It was reported that SI inhibited NF-κB expression in a dose-dependent manner in HEK293 cells, which was stimulated by pro-inflammatory factors TNF-α, IL-1β, and IL-6. Similarly, SI reduced pro-inflammatory factors IL-6 and IL-8 in THP-1 cells induced by lipopolysaccharide [53]. In OGD/R-treated microglial cells, which are often used to evaluate stroke and the consequent inflammatory injury, SI could inhibit proinflammatory cytokines and enzymes, attenuate the nuclear translocation of the NF-κB pathway in BV-2 microglia, and restrain the TLR4/NF-κB pathway or upregulate extracellular heat shock protein 70. These results indicated that SI could effectively inhibit the neuroinflammation induced by stroke [85]. Moreover, SI could attenuate oxidative stress damage by activating the HO-1 pathway and enhancing cellular resistance to hydrogen peroxide-induced oxidative damage [65].

Surprisingly, SI might be used as a potential antitumor agent. Good affinity between SI and C-X-C chemokine receptor type 4 (CXCR4) was observed by affinity detection and SPR ligand screening. The measured affinity constant was 2.94 ± 0.36 μM, indicating that SI might be a potential CXCR4 antagonist that can inhibit the CXCR4-mediated migration of human breast cancer cells [44].

SI showed inhibition capability against cell proliferation. Phthalides from the rhizome of *Cnidium chinensis* were evaluated on smooth muscle cells from a mouse aorta. The order of proliferation–inhibiting efficacy was as follows: senkyunolide L > SH > senkyunolide J > SI > LIG = senkyunolide A > butylidenephthalide, suggesting that SI had an effect to some extent. However, the underlying mechanism is unclear [86].

The BBB permeability of SI was investigated in MDCK-MDR1 cells. The results indicated that SI could enhance cellular transport by downregulating the expression of claudin-5 and zonula occludens-1, two main tight junction proteins that are closely associated with BBB tightness [87]. Additionally, SI decreased the expression of P-glycoprotein (P-gp), which acts as a drug-efflux pump, via the paracellular route to enhance xenobiotics transport [88]. 

**Table 5 molecules-28-03636-t005:** The reported pharmacological effects of SI.

Pharmacological Effect	Cell Line/Animal Model	Action Mechanism	Ref.
Protection of brain	Cerebral ischemia–reperfusion model	Ameliorating neurological injury; reducing cerebral infarct volume; decreasing MDA level; increasing SOD and promoting the expression of p-Erk1/2/t-Erk1/2, c-Nrf2, n-Nrf2, HO-1, and NQO1; and deregulating the expression of Bcl-2, Bax, caspase 3, and caspase 9.	[69]
	SH-SY5Y cells; OGD/R model	Increasing cell viability and decreasing ROS and LDH levels.	[70]
	ICH was induced by intracerebral injection of autologous blood	Ameliorating neurological deficit, brain edema, and neuronal injury; alleviating microglia cell and astrocyting activations; and reducing peripheral immune cell infiltration caused by ICH.	[71]
Protection against septic encephalopathy	Cecal ligation and perforation were used for the sepsis model	Increasing Ngb expression and upregulating the p38 MAPK signal pathway.	[72]
		Improving the survival rate and cognitive dysfunction of sepsis mice; ameliorating systemic inflammatory response; and inhibiting the inflammatory signaling pathway, which includes reducing the phosphorylation levels of JNK, ERK, p38, and p65.	[73]
Protection of liver	Hepatic ischemia–reperfusion; HuCCT1 cells	Decreasing TNF-α, IL-1β, and IL-6; inhibiting P65 NF-κB, MAPK, and MDA; increasing HO-1, SOD, and GSH-Px activities; inhibiting Bax but increasing Bcl-2; and reducing liver tissue apoptosis. It can reduce the damage of HuCCT1 cells, promote Nrf-2 nuclear translocation, and reduce the contents of ROS and MDA in vitro.	[74]
Protection of kidneys	Renal ischemia–reperfusion injury model was established by clipping bilateral renal pedicles; HK2 cells	Protecting renal function and structural integrity; reversing BUN, SCr, and renal pathological damage; inhibiting the secretion of TNF-α and IL-6; reducing ROS production and the expression of endoplasmic reticulum stress-related proteins GRP78 and CHOP.	[75]
Protection of lungs	Lung injury was induced by CLP	Inhibiting the phosphorylation of JNK, ERK, P38, and p65; downregulating the levels of TNF-α, IL-1β, and IL-6 in plasma and lung tissue.	[76]
Protection of blood system	THR (from bovine plasma)	Direct THR inhibitory activity.	[46,77]
	ADP-induced platelet aggregation	Prolonging the PT and APTT activity.	[78]
Protection of vascular system	Zebrafish thrombus model induced by phenylhydrazine	Inhibiting the expression of coagulation factor VII (f7).	[79]
	Erythrocyte deformations induced by ConA	Reducing the deformation and orientation index.	[80]
	Human microvascular endothelial cells	Upregulating placental growth factor.	[82]
Analgesia	Nitroglycerin induced a headache in rats	Reducing NO levels.	[83]
	Nitroglycerin induced a headache in rats	Reducing NO and CGRP levels.	[84]
Anti-inflammatory	Sepsis model induced by CLP; human embryonic kidney 293 cells	Inhibiting the NF-KB signaling pathway.	[53]
	OGD/R model simulates stroke	Suppressing the TLR4/NF-κB pathway by up-regulating Hsp70 dependent on HSF-1.	[85]
Cell transportation	Blood–brain barrier model; MDCK-MDR1 cells	Downregulating the expression of claudin-5 and occlusive zone-1. Increasing the expression of the P-glycoprotein route.	[87]
Antioxidation	HepG2 cells with oxidative damage induced by hydrogen peroxide	Promoting HO-1 expression and inhibiting ROS formation.	[65]
Calcium antagonists	Human embryonic kidney 293 cells; rat cardio myoblast cells (H9C2 cells from ATCC)	Blocking voltage-operated Ca^2+^ channels and ryanodine receptor antagonistic intracellular calcium accumulation.	[42,81]
Antitumor	MCF-7 cells	Direct binding to CXCR4 and inhibition of CXCR4-mediated migration of MCF-7 cells.	[44]

## 7. Pharmacokinetics

Up to now, the pharmacokinetic parameters of SI in rats, mice, rabbits, dogs, and humans have been studied with different administration routes, including intravenous injection, intraperitoneal injection, gavage, etc. The reported pharmacokinetic parameters are summarized in Table 6.

### 7.1. Pharmacokinetic Properties of SI

The pharmacokinetic properties of SI have been studied on animals (mice, rats, and dogs) via different administration routes [35,89,90]. The results indicated that SI would be absorbed rapidly followed by short half-life (<1 h) elimination and acceptable oral bioavailability (>35%) after intragastric administration. SI is widely distributed in tissues and organs in vivo, and the AUC values in descending order were as follows: kidneys > liver > lungs > muscle > brain > heart > thymus > spleen [35].

The pharmacokinetic differences between normal and migrainous rats have been investigated [91]. The results demonstrated that migraines caused some significant changes. For example, the decreased clearance and increased volume of distribution resulted in a several-fold increase in *t*_1/2_ and AUC. The pharmacokinetic parameters of SI were significantly different in normal and migrainous rats, which should be taken into consideration during the design of a clinical dosage regimen for SI.

Similarly, the pharmacokinetic differences of SI and SH in normal and migrainous rats after gavage administration of 70% ethanol extract of *L. chuanxiong* were studied. Compared with normal rats, the absorptions of SI and SH in migraine rats increased significantly, the C_max_ and AUC_(0–t)_ of SI increased by 192% and 184%, while SH increased by 266% and 213%, respectively [43].

Furthermore, the effects of warfarin on the pharmacokinetics of SI in a rat model of biliary drainage following administration of the extract of *L. chuanxiong* were investigated. It was reported that warfarin could significantly increase the *t*_1/2_, *T*_max_, and C_max_ of SI. The result highlights the importance of drug–herb interactions [38].

The metabolic pathways of SI in vivo involve methylation, hydrolysis, and epoxidation of phase I metabolism, as well as glucuronidation and glutathionylation of phase II metabolism. The mainly metabolic pathways in vivo are shown in Figure 5. It was reported that after administration of SI in rats, a total of 18 metabolites were identified in bile, 6 in plasma, and 5 in urine [92]. Ma et al. [40] identified four metabolites of SI in bile, namely, SI-6S-O-β-D-glucuronide, SI-7S-O-β-D-glucuronide, SI-7S-S-glutathione, and SI-7R-S-glutathione. He and colleagues [35] found nine metabolites in rat bile and speculated the metabolic pathways.

### 7.2. Pharmacokinetic Properties of SI Containing Herbal Preparations

Up to now, the pharmacokinetics and metabolism of SI were studied in animals administrated not only with pure SI compound, but also with SI containing herbal preparations.

A total of 25 compounds were detected in plasma after SD rats were gavaged with *L. chuanxiong* decoction, among which 13 were absorbed as prototypes. LIG, the main alkyl phthalide in *L. chuanxiong*, was rapidly absorbed and converted into hydroxyphthalides by phase I metabolism, including SI, SH, senkyunolide F, and senkyunolide G. The absorbed, as well as the generated hydroxyphthalides, were further combined with glutathione or glucuronic acid through phase Ⅱ metabolism [93].

A sequential metabolism approach was developed to study the absorption and metabolism of multiple components in *L. chuanxiong* decoction at different stages of intestinal bacteria, intestinal wall enzymes, and liver metabolism. After enema administration, SI was quickly absorbed as a prototype and stable at each stage of sequential metabolism [94].

SI was used as index component of several herbal preparations, such as Dachuanxiong Pills [95], Shaofu Zhuyu Decoction, and Yigan Powder [62,96]. SI had been detected as one of the main components in plasma and tissues after normal and model animals were administrated. The results confirmed that SI could easily be released from herbal preparations followed by rapid absorption, a short half-time of elimination, and acceptable oral bioavailability in vivo.

Previous studies suggested that there were remarkable differences in SI pharmacokinetics between normal and model animals administrated with SI containing herbal preparations. For example, multi-component pharmacokinetics of the Naomaitong formula was performed in normal and stroke rats. The results indicated that the stroke rats had higher values of AUC_(0–t)_, AUC_(0–∞)_, *t*_1/2_, and MRT_(0–∞)_. The AUC _(0–∞)_ values of SI and LIG were both five times higher than those of the normal rats [97]. Moreover, pharmacokinetics differences were compared after oral administration of Xinshenghua Granules in normal and blood-deficient rats. As a result, a total of 15 components were detected in plasma. However, most of them were eliminated within six hours. The SI values of C_max_, AUC_(0–t),_ and AUC_(0–∞)_ in a blood-deficient rat model were 23%, 32.6%, and 31.6% higher than those of normal rats, respectively [56]. Based on pharmacokinetic experiments in humans and rats, the active phthalides in Xuebijing injection in the treatment of sepsis were determined. A variety of phthalides (SI, SH, senkyunolide G, senkyunolide N, 3-hydroxy-3-N-butylphthalide, etc.) were detected in human and rat plasma, among which both SI and senkyunolide G have significant exposures in plasma [98].

**Table 6 molecules-28-03636-t006:** The reported pharmacokinetic parameters of SI.

Substance	Route of Administration and Dose	Animals/ Model	*C*_max_ (ng/mL)	*T*_max_ (h)	AUC_(0–t)_ (h·ng/mL)	AUC_(0–∞)_ (h·ng/mL)	*T*_1/2_ (h)	MRT_(0–t)_ (h)	CL/F (L h^−1^kg^−1^)	V (L/kg)	Ref.
SI	i.v. (1 mg/kg)	Normal beagle dogs	834.12 ± 89.09	ND	1166.21 ± 189.42	1173.45 ± 134.23	0.62 ± 0.09	0.88 ± 0.05	2.82 ± 0.49	2.34 ± 0.45	[89]
	i.g. (1 mg/kg)		167.45 ± 21.37	0.18 ± 0.02	578.04 ± 123.78	583.25 ± 145.56	0.69 ± 0.11	0.67 ± 0.11	6.21 ± 0.22	4.89 ± 1.06	
	i.g (5 mg/kg)		841.23 ± 120.34	0.22 ± 0.05	2775.98 ± 278.15	2777.42 ± 271.65	0.59 ± 0.18	0.84 ± 0.17	6.52 ± 0.45	5.00 ± 1.73	
	i.g. (50 mg/kg)		7034.12 ± 340.23	0.21 ± 0.02	25,590.58 ± 459.87	25,678.34 ± 501.54	0.75 ± 0.14	1.08 ± 0.09	5.48 ± 0.22	4.67 ± 0.34	
SI	i.g. (18 mg/kg)	Normal SD rats	3310 ± 550	0.22 ± 0.07	10,078,200 ± 894,000	10,152,600 ± 836,400	0.51 ± 0.16	0.77 ± 0.16	ND	4.81 ± 0.1.81	[35]
	i.g. (36 mg/kg)		4420 ± 1520	0.4 ± 0.15	18,046,200 ± 4,118,400	18,420,000 ± 3,910,200	0.66 ± 0.21	0.95 ± 0.13	ND	7.11 ± 3.33	
	i.g. (72 mg/kg)		8960 ± 1080	0.37 ± 0.14	35,776,200 ± 6,591,000	37,138,800 ± 7,714,800	0.75 ± 0.13	0.97 ± 0.11	ND	7.72 ± 1.37	
	i.v. (18 mg/kg)		ND	ND	269,634,000 ± 2,217,000	2,725,560 ± 2,503,800	0.64 ± 0.14	0.57 ± 0.07	ND	2.18 ± 0.37	
	i.v. (36 mg/kg)		ND	ND	47,821,200 ± 5,071,800	48,463,200 ± 4,962,000	0.73 ± 0.12	0.58 ± 0.04	ND	2.84 ± 0.71	
	i.v. (72 mg/kg)		ND	ND	106,417,800 ± 11,481,000	107,790,600 ± 12,061,800	0.66 ± 0.15	0.66 ± 0.12	ND	2.30 ± 0.49	
SI	i.v. (20 mg/kg)	Normal SD rats	13,847,200 ± 2,732,900	ND	7761.1 ± 874.8	7902.5 ± 925.7	0.56 ± 0.13	0.55 ± 0.10	2.56 ± 0.29	ND	[91]
	i.g. (20 mg/kg)	Normal rats	5,236,300 ± 802,800	0.25 ± 0.06	5217.5 ± 1029.5	5458.6 ± 1073.0	0.66 ± 0.19	0.84 ± 0.16	3.78 ± 0.73	ND	
0.23 ± 0.04		Migrainous rats	6,049,400 ± 1,320,700		14,459.1 ± 2130.4	19,477.0 ± 4001.4	5.93 ± 3.61	3.09 ± 0.52	1.06 ± 0.21	ND	
	i.g. (72 mg/kg)	Normal rats	22,071,900 ± 3,456,100	0.38 ± 0.11	21,480.2 ± 3003.1	21,953.0 ± 3162.1	0.52 ± 0.12	0.80 ± 0.09	3.34 ± 0.54	ND	
		Migrainous rats	23,599,100 ± 8,052,700	0.416667	45,177.0 ± 14,366.9	61,810.5 ± 12,086.8	10.44 ± 11.64	2.60 ± 0.41	1.20 ± 0.25		
SI	i.g. (32 g/kg)	Normal rats	3900 ± 900	0.4 ± 0.1	16,600 ± 500	17,800 ± 470	6.3 ± 2.2	ND	ND		[43]
		Migrainous rats	11,400 ± 3600	0.4 ± 0.1	47,200 ± 19,200	49,600 ± 20,700	4.8 ± 2.4	ND	ND		
*L. chuanxiong*	i.g. (10 g/kg)	Normal SD rats	92.33 ± 19.69	0.25 ± 0.00	842.74 ± 16.13	ND	1.03 ± 0.35	0.78 ± 0.04	ND	ND	[38]
		SD rats with biliary drainage	371.49 ± 94.10	0.28 ± 0.09	1,822,573.2 ± 638,426.4	ND	1.18 ± 0.45	1.20 ± 0.23	ND	ND	
*L. chuanxiong* and warfarin	i.g. (10 g/kg)	Normal SD rats	322.36 ± 213.54	0.28 ± 0.09	1,615,385.4 ± 756,679.8	ND	1.45 ± 0.34	1.35 ± 0.19	ND	ND	
		SD rats with biliary drainage	208.85 ± 80.64	0.47 ± 0.09	1,292,366.4 ± 586,936.8	ND	1.43 ± 0.35	1.26 ± 0.27	ND	ND	
Shaofu Zhuyu Decoction	i.g. (0.5 mg/kg)	Normal beagle dogs	92.8 ± 4.9	0.3	324.9 ± 38.3	ND	1.3 ± 0.1	ND	ND		[62]
YiGan San	i.g. (9.1 g/kg)	Normal SD rats	136.02 ± 39.64	0.94 ± 0.83	718.29 ± 137.86	898.76 ± 265.79	1.52 ± 0.03	5.66 ± 0.66	223.72 ± 56.06	ND	[96]
Xiaoyao Powder	i.g. (4 g/kg)	Normal SD rats	403.26 ± 201.00	0.36 ± 0.13	1582.38 ± 985.86	1616.46 ± 967.01	4.75 ± 2.78	5.31 ± 0.72	ND	ND	[55]
Naodesheng	i.g. (4 g/kg)	Wistar rats	6990 ± 3240	0.5 ± 0.15	9960 ± 2390	10,160 ± 2510	1.66 ± 0.63	1.91 ± 0.48		ND	[99]
XueBiJing injection	i.v. (100 mL/day)	Human	313 ± 57	ND	ND	571 ± 115	0.87 ± 0.09	1.73 ± 0.14	ND		[98]
SI	i.v. (104 mg/kg)	Normal KM mice	84.21	0.03	163,640,400	164,328,000	0.53	0.53		1.75	[90]
	i.g. (104 mg/ kg)		12.31	0.33	52,681,200	53,545,200	0.67	0.96		6.74	[57]
Xian-Xiong-Gu-Kang	i.g. (1.5 mL/100g)	SD rats with osteoarthritis model	77.15 ± 22.84	0.63 ± 0.14	441.06 ± 173.27	ND	6.26 ± 1.09	6.23 ± 1.15	ND	ND	
NaoMai Tong	i.g. (6 g/kg)	Normal SD rats	16.04 ± 9.43	0.50 ± 0.25	73.55 ± 45.87	90.28 ± 32.74	11.12 ± 35.66	20.93 ± 47.99	ND	ND	[97]
		SD rats with stroke-afflicted	267.38 ± 164.02	0.45 ± 0.33	454.76 ± 129.26	487.84 ± 132.21	7.48 ± 4.44	6.05 ± 2.95	ND	ND	
Xin-Sheng-Hua Granule	i.g. (9.86 g/kg)	Normal SD rats	86.88 ± 13.12	0.42 ± 0.00	144.41 ± 17.38	152.45 ± 17.21	2.54 ± 0.93	2.99 ± 0.26	ND	ND	[56]
		SD rats with blood deficiency	106.09 ± 17.09	0.67 ± 0.00	191.56 ± 25.86	200.78 ± 25.15	0.67 ± 0.00	2.41 ± 0.41	ND	ND	

Abbreviations: AUC, area under the concentration–time curve; C_max_, maximum plasma concentration; MRT, mean residence time; i.g., gavage administration; i.v., intravenous administration; ND, not determined.

## 8. Conclusions and Future Perspective

The structural variety and biological correspondence of natural products have provided beneficial enlightenment for new drug discovery and development. A valid strategy is to screen potential candidates from traditional herbal medicine with historically proven effects, such as morphine from poppy, artemisinin from sweet wormwood, and salicylic acid from willow bark. Unfortunately, many natural products, despite their significant bioactivity, fail to meet the requirements of qualified drug candidates due to unsatisfactory safety, stability, solubility, bioavailability, or other druggable deficiencies. In this case, their natural or modified derivates are often researched to discover potential substitutes with superior druggable properties and comparable bioactivities.

Despite their low bioavailability, LIG and NBP present outstanding neuroprotective effects. SI is an oxidation product and an in vivo metabolite of LIG. Compared with LIG, SI is more chemically stable, easily soluble in water, and presents significantly better bioavailability. Furthermore, SI can permeate the BBB, which means it can access the brain’s disease focus directly. These properties make SI a potentially useful medicinal compound.

Nevertheless, further studies need to be performed before SI can be considered a candidate to comprehensively assess its druggability. First, it is necessary to develop a preparation method that can obtain large quantities of SI at a low cost, thus providing substantial material for efficacy assessment, safety studies, and new drug development. Second, the efficacy evaluation and mechanism clarification of SI are still insufficient compared to LIG. In particular, in vivo comparative studies of SI with similar drugs or components, such as NBP and LIG, are needed to address the effectiveness and potential advantages of SI. Third, a structure–activity comparison between SI and similar phthalides would be useful. SI is a product of dihydroxylation of the six and seven double bonds of LIG. The introduction of o-dihydroxyl significantly improves the water solubility of the molecule while leaving BBB transmissibility unchanged. The structural properties and mechanisms of the transmissibility of SI across the BBB deserve more investigation, which may provide valuable references for subsequent structural modifications and the design of other drug molecules.

## Figures and Tables

**Figure 1 molecules-28-03636-f001:**
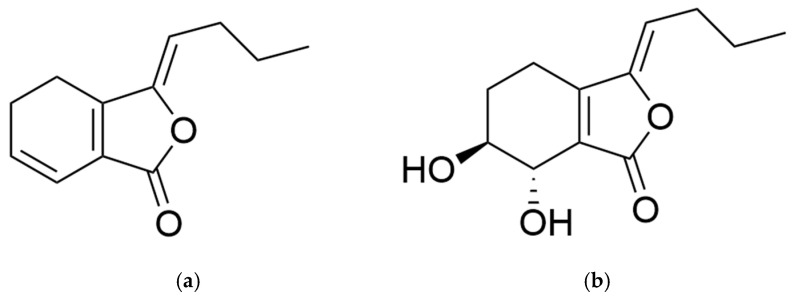
Chemical structure of (**a**) ligustilide and (**b**) senkyunolide I.

**Figure 2 molecules-28-03636-f002:**
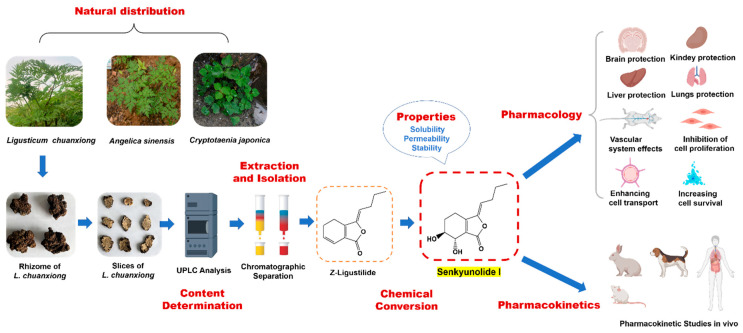
An illustrated summary of senkyunolide I.

**Figure 3 molecules-28-03636-f003:**
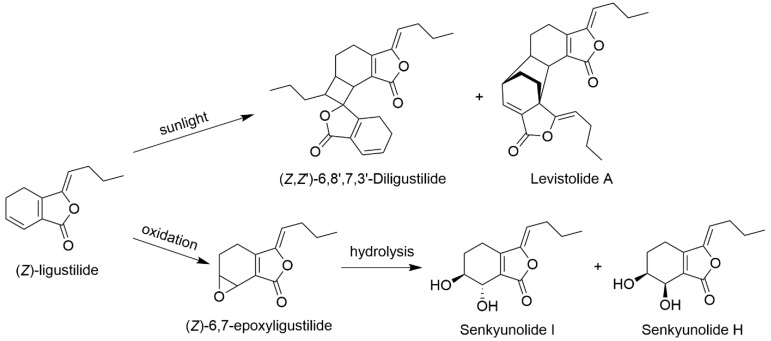
Chemical transformation of ligustilide.

**Figure 4 molecules-28-03636-f004:**
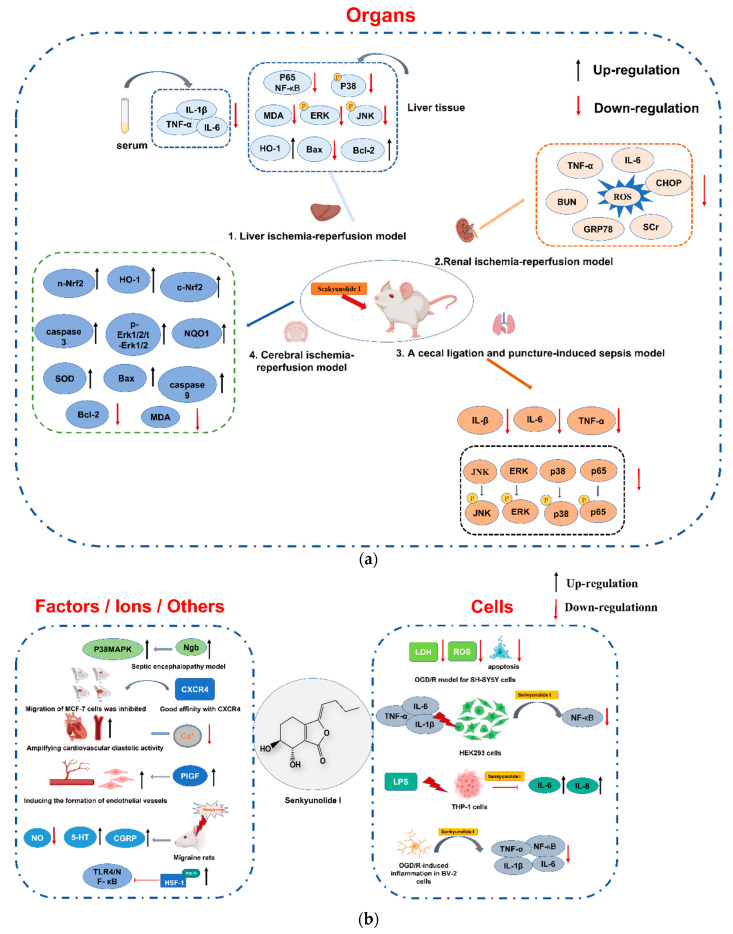
Major pharmacological effects. (**a**) Pharmacological effect on organs; (**b**) Pharmacological effect on cells/factors/icons/others.

**Figure 5 molecules-28-03636-f005:**
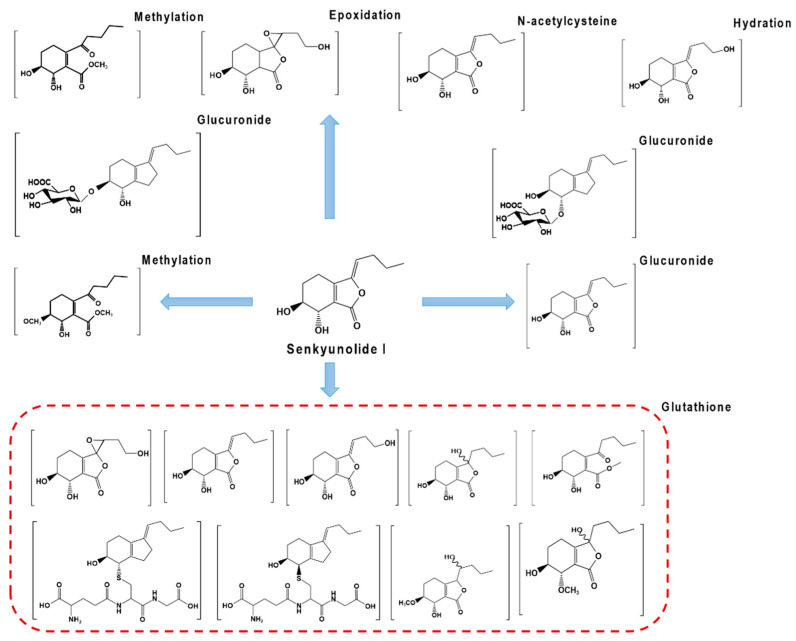
Main metabolic pathways of SI in vivo. The main metabolic reactions and proposed metabolites of SI, including G-SH: glutathione; glucuronide; N-acetylcysteine; methylation; epoxidation; hydration. Figure 5 is referenced by [35,40,92].

**Table 1 molecules-28-03636-t001:** Analytical methods and their main parameters.

Sample	Detection Method	Mobile Phase	Column	Ref.
SI	HPLC	0.05% phosphate–methanol	Apollo C18 (150 × 4.6 mm)	[39]
SI	LC-MS/MS	0.1% formic acid–acetonitrile	Acquity BEH C18 column (2.1 × 50 mm, 1.7 μm)	[27]
SI	UPLC–MS/MS	0.1% formic acid–acetonitrile	Acquity BEH C18 (2.1 × 100 mm, 1.7 μm)	[38]
SI	UPLC/Q-TOF-MS	0.1% formic acid–acetonitrile	Grace C18 (4.6 × 150 mm,5 μm); Acquity HSS T3 (2.1 × 100 mm, 1.8 μm)	[40]
Root of *A. sinensis*	HPLC	1% acetic acid–acetonitrile	Prevail 18 (4.6 × 250 mm, 4 μm)	[41]
Root of *A. sinensis*	UHPLC-QTOF-MS/MS	water–acetonitrile (both containing 0.1% formic acid)	CORTECS C18 (2.1 × 100 mm, 1.6 μm)	[42]
Rhizome of *L. chuanxiong*	LC-MS	water–methanol	Kromasil C18 (250 × 4.6 mm, 5 μm)	[43]
Rhizome of *L. chuanxiong*	HPLC-DAD-MS	0.5% acetic acid–acetonitrile	Alltima C18 (4.6 × 250 mm, 5 μm)	[7]
Rhizome of *L. chuanxiong*	HPLC-ESI-MS/MS	0.5% acetic acid–acetonitrile	Sapphire C18 (4.6 × 250 mm, 5 µm)	[35]
Rhizome of *L. chuanxiong*	UPLC-QTOF-MS	0.1% formic acid–acetonitrile	X Select HSS T3 (2.1 × 100 mm, 2.5 μm)	[44]
Rhizome of *L. chuanxiong*	HPLC-ESI-Q-TOF-MS/MS	0.1% formic acid–methanol	Kromasil-C18 (4.6 × 250 mm, 5 µm)	[45]
Rhizome of *L. chuanxiong*	HPLC-DAD; UPLC-QTOF-MS	0.1% formic acid–methanol; 0.1% formic acid–methanol	ZORBAX SB-C18 (4.6 × 250 mm, 5 µm); Acquity BEH C18 (100 × 2.1 mm, 1.7 μm)	[46]
Rhizome of *L. chuanxiong*	GC-MS; TLC; HPLC-DAD; HPLC-MS	water–methanol	Zorbax SB-C18 (250 × 4.5 mm, 5 µm)	[47]
Xue-Fu-Zhu-Yu decoction	HPLC-ESI-MS	0.08% acetic acid–methanol	Zorbax SB-C18 (4.6 × 250 mm, 5 µm)	[48]
Siwu decoction	UPLC-QTOF/MS/MS; HPLC-DAD	0.1% formic acid–acetonitrile	Acquity BEH C18 (100 × 2.1 mm, 1.7 μm)	[49]
Danggui Buxue decoction	LC-MS; HPLC-DAD-ELSD	0.1% formic acid–acetonitrile	Zorbax C18 (250 × 4.6 mm, 5 μm)	[50]
Danggui oral solution	LC-DAD-APCI-MS	water–methanol	Hypersil-Keystone C18 (150 × 2.1 mm, 5 μm)	[51]
Guanxinning injection	HPLC-DAD-ESI-MS	0.08% formic acid–methanol	Ultimate XB-C18 (250 × 4.6 mm, 5 μm)	[52]
Xuebijing injection	UPLC-Q/TOF	(formic acid–acetonitrile–methanol, 0.5:60:40)–(formic acid–water, 0.5:100)	Acquity BEH C18 column (2.1 × 100 mm, 1.7 μm)	[53]
Danggui-Shaoyao powder	HPLC-DAD-ESI-MS/MS	0.01% formic acid–acetonitrile	Alltima C18 (250 × 4.6 mm, 5 μm)	[54]
Xiaoyao powder	UPLC-MS/MS	(0.1% formic acid)–acetonitrile	Acquity BEH C18 (50 × 2.1 mm, 1.7 µm)	[55]
Xin-Sheng-Hua granule	UPLC-TQ-MS/MS	(0.1% formic acid)–acetonitrile	Hypersil GOLD (100 × 3 mm, 1.9 μm)	[56]
Xian-Xiong-Gu-Kang	LC-MS/MS	(0.05% formic acid)–acetonitrile	Acquity BEH C18 (2.1 × 100 mm, 1.7 μm)	[57]
Bu-Zhong-Yi-Qi Wan	SPE-HPLC-DAD-ELSD	water–acetonitrile	Spursil C18 (250 × 4.6 mm, 5 μm)	[58]

**Table 2 molecules-28-03636-t002:** Contents of SI in medicinal materials.

Medicinal Material	Content	Ref.
*Angelica sinensis* root	0.137~0.505 mg/g	[59]
*Angelica sinensis* root	0.149~1.006 mg/g	[60]
*Angelica sinensis* root	0.276~0.296 mg/g	[42]
*Ligusticum chuanxiong* rhizome/*Angelica sinensis* root	0.065~2.158 mg/g	[61]
*Ligusticum chuanxiong* rhizome	10.9 mg/g	[7]
Chuanxiong Dispensing Granules	2.08~6.07 mg/g	[23]

**Table 3 molecules-28-03636-t003:** Contents of SI in preparations.

Preparation Name	Content	Ref.
Xiangfu-Siwu decoction	0.02 mg/g	[49]
Taohong-Siwu decoction	0.03 mg/g	[49]
Qinlian-Siwu decoction	0.08 mg/g	[49]
Siwu decoction	0.18 mg/g	[49]
Shaofu-Zhuyu decoction	0.21 mg/g	[49]
Shaofu-Zhuyu decoction	0.4 mg/g	[62]
Wuji Baifeng pills	0.069 mg/g	[60]
Xiaoyao pills	0.098 mg/g	[60]
Bazhen Yimu pills	0.128 mg/g	[60]
Danggui Futongning pills	0.142 mg/g	[60]
Bu-Zhong-Yi-Qi pills	0.152 mg/g	[60]
Aifu Nuangong pills	0.163 mg/g	[60]
Danggui Kushen pills	0.169 mg/g	[60]
Concentrated Danggui pills	0.423 mg/g	[60]
Bu-Zhong-Yi-Qi pills	0.064~0.136 mg/g	[58]
Fuke Tiaojing tablets	0.261 mg/g	[60]
Niuhuang Shangqing tablets	0.292 mg/g	[60]
Tiaojing Zhitong tablets	2.206 mg/g	[60]
Xin-Sheng-Hua granules	0.003 mg/ml	[56]
Yangxue Qingnao granules	0.323 mg/g	[60]
Danggui-Shaoyao powder	0.9 mg/g	[54]
Xiaoyao powder	1.45 mg/g	[55]
Guanxinning injection	0~0.478 mg/ml	[52]
Xian-Xiong-Gu-Kang	0.4 mg/ml	[57]

## Data Availability

Not applicable.

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
