# Peer review of "Senkyunolide I: A Review of Its Phytochemistry, Pharmacology, Pharmacokinetics, and Drug-Likeness"

_molecules, 2023, doi:10.3390/molecules28083636_

Round 1

Reviewer 1 Report

I have reviewed the manuscript entitled "Senkyunolide IA review of its phytochemistry, pharmacology, pharmacokinetics, and drug-likeness". It needs minor revisions before publication in Molecules. Detailed remarks about the paper are as follows.

1. Please provided an relevant figure to better demonstrate the “Chemical Transformation of LIG

2. In the part of “Metabolic Transformation of LIG”, does there have any information pointed the key enzyme to transfer LIG to SI?

3. In subsection 4, authors need to add more discussion information on the content of SI, rather than just list all about analysis information in the literature.

Author Response

                                Response to Reviewer 1 Comments

Point 1

Please provided an relevant figure to better demonstrate the “Chemical Transformation of LIG”

Response 1: Thank you for your valuable suggestion. We have supplemented the transforming diagram from LIG to SI in Figure 3 and added it to the corresponding position in the text.

         Figure 3. Chemical transformation of ligustilide.

Point 2

In the part of “Metabolic Transformation of LIG”, does there have any information pointed the key enzyme to transfer LIG to SI?

Response 2: In this section, based on our understanding of relevant literature, there is no clear indication of the key enzyme involved in the metabolic conversion of ligustilide to SI. There have been literature studies on the enzyme kinetics of ligustilide in rat liver microsomes, suggesting that CYP3A4, CYP2C9, and CYP1A2 are the main metabolic enzymes involved in LIG metabolism. However, the authors did not point out that these enzymes are the characteristic key enzymes for the metabolism of LIG conversion to SI. The comments above has been added in the section of the revised manuscription.

Point 3

In subsection 4, authors need to add more discussion information on the content of SI, rather than just list all about analysis information in the literature.

Response 3: Many thanks for your advice. There were indeed some omissions in this section, and now we have added more discussion information on the specific content as 4.2 “Content of Medical Material and Preparation”, the details are as follows:

"Among commonly used TCM, SI exists limitedly in A. sinensis and L. chuanxiong. Table 2 indicates that the maximum content of SI in A. sinensis is 1mg/g, while it is up to more than 10mg/g in L. chuanxiong. The reason is presumably that LIG in L. chuanxiong is in higher level, and might produce more SI, compared with that in A. sinensis. In addition, SI concentrations in Chuanxiong dispensing granules range from 2.08 to 6.07 mg/g. The relative high content might be attributed to its good water solubility, or accelerated transformation from LIG during decocting, concentrating or drying processes. Table 3 summarizes quantitative analysis of SI in multiple compound preparations containing L. chuanxiong rhizome and A. sinensis root. The result shows a large fluctuation from 0.02 to 2.206mg/g, suggesting that SI content is probably influenced by material quality, formulation and preparation technology."

Reviewer 2 Report

The manuscript aims to give an overview of various potential pharmacological activities of a natural phthalide, called Senkyunolide I (SI). The English language should definitely be improved on first place, before further submission decisions; in the present form, a lot of the sentences are ambiguous.  

Page 1; Lines 29, 35

Please, include a citation after every mentioned pharmacological activity.

Line 39

“indication of mild and moderate acute ischemic stroke … “, please provide citation

Table 2 and 3, Line 4, Page 8

It is not clear what are the purposes of these two tables and how they are connected with the text.

Line 12 and 13

“hydroxy-propyl-β-cyclodextrin complex [92, 93]…”, such data should be provided in the introduction or discussion, not in the conclusion section. The information that you state in the conclusion section should be a summary of information presented earlier in the manuscript.

The manuscript would benefit if there is more connection between different paragraphs in the text so that the text is more structured.

In summary, before further considerations on the submission, the English language should be improved significantly, on first place.

Author Response

Response to Reviewer 2 Comments

Point 1:

The manuscript aims to give an overview of various potential pharmacological activities of a natural phthalide, called Senkyunolide I (SI). The English language should definitely be improved on first place, before further submission decisions; in the present form, a lot of the sentences are ambiguous.

Response 1:We are sorry for our deficiencies in the writing quality.  We have tried our best to polish the entire text and modify mistakes and unclear expressions under the help of a native English speaker.

Point 2

Page 1; Lines 29, 35

Please, include a citation after every mentioned pharmacological activity.

Response 2:We have rechecked the entire text for any omissions, and have supplemented the missed references in the corresponding positions in the pharmacological effects section, i.e., [2]、[3]、[4]、[5]、[65].

Point 3:

Line 39

“indication of mild and moderate acute ischemic stroke ... ”, please provide citation

Response 3We have supplemented the reference in the corresponding position in the text. The corresponding reference is [9].

Point 4:

Table 2 and 3, Line 4, Page 8

It is not clear what are the purposes of these two tables and how they are connected with the text.

Response 4Thank you for your suggestion to help us to further improve section 4. We have noticed this issue and explained the reasons for listing the content of SI in medicinal materials or preparations. We also further summarized the content in Tables 2 and 3. The details are as follows:

Among commonly used TCM, SI exists limitedly in A. sinensis and L. chuanxiong. Table 2 indicates that the maximum content of SI in A. sinensis is 1mg/g, while it is up to more than 10mg/g in L. chuanxiong. The reason is presumably that LIG in L. chuanxiong is in higher level, and might produce more SI, compared with that in A. sinensis. In addition, SI concentrations in Chuanxiong dispensing granules range from 2.08 to 6.07 mg/g. The relative high content might be attributed to its good water solubility, or accelerated transformation from LIG during decocting, concentrating or drying processes. Table 3 summarizes quantitative analysis of SI in multiple compound preparations containing L. chuanxiong rhizome and A. sinensis root. The result shows a large fluctuation from 0.02 to 2.206mg/g, suggesting that SI content is probably influenced by material quality, formulation and preparation technology.

Point 5

Line 12 and 13

“hydroxy-propyl-B-cyclodextrin complex [92, 93...", such data should be provided in the introduction or discussion, not in the conclusion section. The information that you state in the conclusion section should be a summary of information presented earlier in the manuscript.

Response 5: The description “Researchers achieved some improvement by preparing it into nano-emulsion and hydroxypropyl-β-cyclodextrin complex [92, 93], but specific technique is need and manufacture cost is high.” has been moved to introduction part with corresponding modification.

Point 6:

The manuscript would benefit if there is more connection between different paragraphs in the text so that the text is more structured.

In summary, before further considerations on the submission, the English language should be improved significantly, on first place.

Response 6: We are sorry for our deficiencies in the writing. We have made great efforts to improve the logical connection and language quality of the manuscript. Thanks for your kind patience.

Round 2

Reviewer 2 Report

The authors have addressed most of the previous issues and the quality of the manuscript has improved, however, there are still some typos throughout the text. Also, please check the correct formatting of the whole manuscript, text, tables, figures.

Please improve the image quality (higher resolution) of Figures 2-4. The text is totally incomprehensible there.

Figure 4

"cells/factors/icons/others" does not seem to give any information.

Line 104

"SI (8, 16, 32 mg/kg) and (18, 36, 72 mg/kg)"

Table 6

Some text seem to be displaced outside of page dimensions.

"8. Conclusion and Future perspective"

"Future perspective" does no need to be with a capital letter.

After minor revision, the manuscript would be suitable for publication.

Author Response

                            Response to Reviewer 2 Comments

Point 1: The authors have addressed most of the previous issues and the quality of the manuscript has improved, however, there are still some typos throughout the text. Also, please check the correct formatting of the whole manuscript, text, tables, figures.

Response 1: Thank you for your suggestion to help us to further improve our manuscript. Once again, we have rechecked the typos throughout the text and corrected the formatting of the whole manuscript, text, tables, figures. As a result, we made adjustments to the table format and minor modifications were made to some of the figures, such as figure 2, figure 4, and figure 5.

Point 2:Please improve the image quality (higher resolution) of Figures 2-4. The text is totally incomprehensible there.

Response 2: We have replaced the image with a higher resolution.

Point 3:

Figure 4

"cells/factors/icons/others" does not seem to give any information.

Response 3: We have divided the pharmacological effects of SI into three parts, they are organs, factors/ions/others, and cells. Different colored arrow () symbols represent different pharmacological mechanisms (up-regulation or down-regulation). With your suggestion, we have made some adjustments to the content of Figure 4, hoping to facilitate the understanding of SI's pharmacological mechanisms.

Point 4:

Line 104

"SI (8, 16, 32 mg/kg) and (18, 36, 72 mg/kg)"

Response 4: Due to our expression mistake, the sentence is difficult to understand. We have made modifications to the corresponding positions in the text “6.4. Other Pharmacological Effects”, the details are as follows:

“Previous studies demonstrated that SI has a unique effect on anti-migraines effect. Kunming mice and SD rats were given different doses of SI (8, 16, 32 mg/kg) and (18, 36, 72 mg/kg) by gavage, respectively.” repalced by “The analgesic effect of SI was evaluated by acetic-acid-induced writhing test on Kunming mice (8, 16, 32 mg/kg, ig), and the anti-migraines activity was tested by nitroglycerin-induced headaches in SD rats (18, 36, 72mg/kg, ig).”

Point 5:

Table 6

Some text seem to be displaced outside of page dimensions.

Response 5: Thank you for your attention. The format of table 6 may have changed due to modifications to the article. What you are seeing now is our adjusted table. And then, we also rechecked the entire tables.

Point 6:

"8. Conclusion and Future perspective"

"Future perspective" does no need to be with a capital letter.

Response 6: We have changed the capital letter.

After minor revision, the manuscript would be suitable for publication.

Reply: We deeply appreciate your precious patience and kind help to our manuscript processing.